# Optimal Operation of a Residential Battery Energy Storage System in a Time-of-Use Pricing Environment

**Charalampos Galatsopoulos** [1,*]**, Simira Papadopoulou** [1,2] **, Chrysovalantou Ziogou** [1]**, Dimitris Trigkas** [1] **and Spyros Voutetakis** [1]

[1] Chemical Process and Energy Resources Institute (CPERI), Center for Research and Technology Hellas (CERTH), Thermi, P.O. Box 60361, 57001 Thessaloniki, Greece; shmira@certh.gr (S.P.); cziogou@certh.gr (C.Z.); dtrigas@certh.gr (D.T.); paris@certh.gr (S.V.)

[2] Department of Industrial Engineering and Management, International Hellenic University (IHU), Thermi, 57001 Thessaloniki, Greece; shmira@ihu.gr

\* Correspondence: cgalatso@certh.gr

**Abstract:** Premature ageing of lithium-ion battery energy storage systems (BESS) is a common problem in applications with or without renewable energy sources (RES) in the household sector. It can result to significant issues for such systems such as inability of the system to cover load demand for a long period of time. Consequently, the necessity of limiting the degradation effects at a BESS leads to the development and application of energy management strategies (EMS). In this work, EMSs are proposed in order to define optimal operation of a BESS without RES under time-of-use (ToU) tariff conditions. The objective of the developed EMSs is to reduce the capacity loss at the BESS in order to extend its lifetime expectancy and therefore increase the economic profit in the long-term. The EMSs utilize a widely used battery mathematical model which is experimentally validated for a specific BESS and a battery degradation mathematical model from the literature. Indicative simulation results of the proposed strategies are presented. The outcomes of these simulated scenarios illustrate that the objectives are achieved. The BESS operates efficiently by preventing premature ageing and ensuring higher economic profit at the long term.

**Keywords:** battery energy storage system; battery ageing estimation; energy management; dynamic optimization

## 1. Introduction

In recent years, the excessive global increment of energy consumption and the issue of environmental pollution has led to the shift towards a green economy. The scientific community has put a significant amount of effort into improving the manufacturing quality and performance of the devices which are utilized in renewable energy systems (such as photovoltaics, wind generators and batteries). Nevertheless, the implementation of control strategies in order to guarantee the optimization of the system's performance remains crucial.

In that context, battery energy storage systems (BESS) are going to play a key role in the future in many sectors [1]. However, BESSs have not proven yet notable performances in the household sector [1]. In [2,3] it is analyzed that the application of a photovoltaic (PV) battery system in a household is a significantly challenging issue regarding its sustainability. This is mainly due to the high investment costs for the development of a BESS. Furthermore, in most countries single rate (SR) energy tariffs and time-of-use (ToU) energy tariffs are used. This implies that applying a BESS without renewable energy sources (RES) is at the first case (SR) infeasible and at the second case (ToU) extremely challenging. Additionally, despite the continuous decrease of the lithium-ion battery costs and the consecutive

improvement of the batteries' performance, the value for money relation still cannot be considered ideal. In order to overcome these difficulties, various economic and technical studies [4,5] have been performed over the past years which have proved that it remains unclear whether a residential PV battery system can operate profitably. Moreover, in [6] it is proved that achieving 100% autonomy in a household with a PV battery system is not a realistic scenario in most countries in Europe without oversizing the PV or the BESS. In addition, in [7] the importance of developing management strategies for analyzing the performance of the PV battery system is highlighted. Various scenarios are analyzed such as operation of the system without PV and operation without BESS. Another remarkable example of techno-economic studies is the work presented in [8], where a decision support tool for a BESS with integrated photovoltaic is presented. This tool aims to define the optimal sizing of BESS and optimal operation scheduling by taking under consideration various parameters such as weather data, electricity pricing environment and BESS specifications/costs. In addition, plenty methodologies for calculating the levelized cost of energy (LCOE) have been developed. A methodology which intends to minimize the LCOE by defining optimal photovoltaic (PV) rated power and BESS capacity is analyzed in [9]. Methodologies for calculating LCOE, economic and technical studies are implemented so as to define the optimal PV rated power and BESS capacity by taking into account specific conditions like building dimensions, geographical location, climate, energy consumption per season and energy market. Therefore, they cannot be considered suitable for universal use. All the aforementioned factors prove the difficulty of ensuring sustainability of BESSs in building applications and hence they evince the requirement of applying efficient energy management strategies (EMS).

A BESS can be used in a residential building with RES or without in case of the existence of a ToU tariff or real time pricing (RTP) tariff. Nonetheless, efficient control of BESSs plays a significant role in achieving sustainability regardless the existence of RES [10,11]. A non-constrained operation will result in a long operation time for the BESS and/or high operating currents and thus to significant degradation effects. In [12] the economic impact of battery ageing in a residential PV battery system is highlighted. Moreover, in [13] a methodology is presented for preventing battery degradation in a residential BESS by using forecast-based operating strategies. This is achieved by storing in the BESS only the amount of energy which is estimated to be needed during the night. This yields lower depth of discharges and hence, reduced capacity losses for the BESS. However, in case of inaccurate estimations for the needed energy, decreased economic profit might be observed. Furthermore, in [14] is proposed a cooperative energy management between a utility and households with PV battery systems. The energy management is examined under RTP tariff and aims to operate the BESS at minimum cost for each household.

The objective of this work is to develop EMSs for a residential BESS which operates in a ToU tariff scenario. In this work the operation of the BESS is examined without a RES. The novelty in this paper comes in the achievement of the maximum possible economic profit for the consumer indirectly by decreasing the degradation effects to the BESS and hence, extending its lifetime expectancy. The goal is to prove that it is possible to secure increased economic profit for the consumer even in a ToU tariff and with the absence of a RES. In order to achieve that, the EMSs utilize a widely used battery mathematical model which is experimentally validated for a specific BESS and a battery degradation mathematical model from the literature. Furthermore, dynamic optimization methodologies which take under consideration forecasted profiles for the household's energy consumption are developed.

This paper is organized as follows: Section 2 presents a specific residential BESS and a seasonal ToU tariff in which the EMSs are applied. Section 3 presents the kinetic battery model (KiBaM) and a battery degradation model for lithium iron phosphate batteries. Section 4 outlines the development of EMSs for application in residential BESS without RES at the seasonal ToU tariff. Section 5 demonstrates the operation of the aforementioned premature-ageing preventing strategies and analyzes their performance. Finally, at Section 6 the concluding remarks of this work are presented.

## 2. Residential BESS without RES in ToU Tariff

The residential BESS to be considered consists of 15 lithium iron phosphate battery cells (Model: GBS-LFP100Ah-A, Zhejiang GBS Energy Co., Ltd., Shanghai, China) connected in series. The nominal voltage of the GBS-LFP100Ah-A battery is 3.2–3.4 V and the nominal capacity 100 Ah. Consequently, the nominal voltage of the 15-lithium iron phosphate battery-stack is 48–51 V, the capacity is 100 Ah and the overall power capacity is 4.8–5.1 kW.

The application which is going to be explored considers the operation of the aforementioned BESS without RES at a household in Greece where the energy pricing environment is ToU. The performance of the BESS is explored by taking into account different household power consumption profiles according to the four seasons of the year. Moreover, it is taken under consideration that the ToU tariff in Greece is not identical through all the year. During spring and summer, the tariff is 0.078 €/kWh from 11 p.m. to 7 a.m. and 0.11 €/kWh from 7 a.m. to 11 p.m. (Figure 1a). On the contrary, during autumn and winter the tariff is 0.078 €/kWh from 2 a.m. to 8 a.m. and from 3 p.m. to 5 p.m. while the tariff is 0.11 €/kWh from 8 a.m. to 3 p.m. and from 5 p.m. to 2 a.m. (Figure 1b).

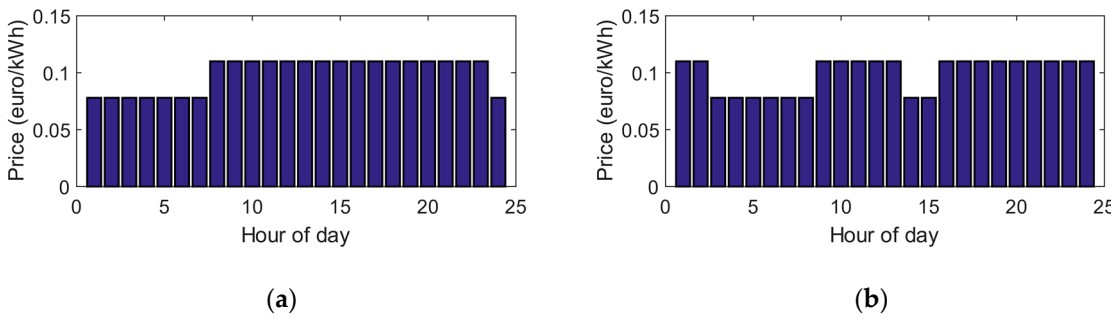

(**a**)                                                    (**b**)

**Figure 1.** ToU tariffs: (**a**) spring and summer tariff; (**b**) autumn and winter tariff.

Owing to the absence of RES, the BESS is charged from the grid when the tariff is 0.078 €/kWh and discharges to the house when the tariff is 0.11 €/kWh. BESS supplies power to the household in parallel operation with the grid in order to guarantee uninterrupted supply and ensure the absence of dead time during transition (Figure 2). Furthermore, the parallel operation of the BESS with the grid secures that the excessively high instantaneous loads which cannot be covered by the BESS itself due to current limitations will be covered.

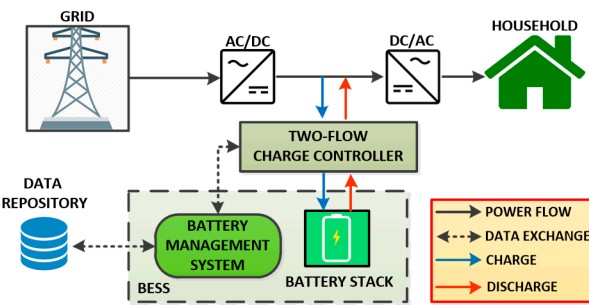

**Figure 2.** Residential BESS without RES.

The BESS comprises the 15 lithium-ion battery stack and the battery management system (BMS) which contains the EMSs. BMS retrieves day-ahead power consumption profile and ToU tariffs from the data repository. Thereafter, predicts and applies the optimal charging/discharging operation schedule to the battery stack in order to prevent premature degradation of the batteries and secure high economic profit. Over and above, it supervises the whole operation of the system, repossesses updated power consumption profiles and updates the optimal charging/discharging operation schedule.

### 3. Battery Mathematical Models

The KiBaM model [15] is presented and validated for experimental data of the 15 lithium-iron phosphate battery-stack described in Section 2. In addition, a battery degradation model for lithium iron phosphate batteries is presented.

#### 3.1. KiBaM Model

The KiBaM model is well known globally for its high accuracy and adaptability to various types of batteries. The operation of the battery is described as two tanks that are connected with a regulation valve. The first tank represents the charge that is available for use and the second one the charge that is chemically bound. The KiBaM model is described as:

$$q_1 = q_{1,0}e^{-kt} + \frac{(q_0 kc - I_{ac})\left(1 - e^{-kt}\right)}{k} - \frac{I_{ac}c\left(kt - 1 + e^{-kt}\right)}{k} \tag{1}$$

$$q_2 = q_{2,0}e^{-kt} + q_0(1 - c)\left(1 - e^{-kt}\right) - \frac{I_{ac}(1 - c)\left(kt - 1 + e^{-kt}\right)}{k} \tag{2}$$

where $q_1$, $q_2$ represent the available and chemically bound charge in Ah, $q_{1,0}$ and $q_{2,0}$ the initial values of $q_1$, $q_2$ in Ah, $k$ and $c$ are estimated based on manufacturer data, $t$ symbolizes the time in h and $I_{ac}$ the charging/discharging current in A.

The maximum discharging current is calculated by (3) which is obtained by setting $q_1 = 0$ in (1).

$$I_{d,\max} = \frac{kq_{1,0}e^{-kt} + q_0 kc\left(1 - e^{-kt}\right)}{1 - e^{-kt} + c(kt - 1 + e^{-kt})} \tag{3}$$

Likewise, the maximum charging current is given by (4) which is obtained by setting $q_1 = cq_{\max}$ in (1).

$$I_{c,\max} = \frac{-kcq_{\max} + kq_{1,0}e^{-kt} + q_0 kc\left(1 - e^{-kt}\right)}{1 - e^{-kt} + c(kt - 1 + e^{-kt})} \tag{4}$$

The parameter $q_{\max}$ denotes the combined volume of the tanks and is given by (5).

$$q_{\max} = \frac{q_t\left[\left(1 - e^{-kt}\right)(1 - c) + kct\right]}{kct} \tag{5}$$

where $q_t$ symbolizes the capacity that corresponds to current $I$ at time $t$.

The battery's voltage is calculated as:

$$V_{ac} = E_{ac} - I_{ac}R_o \tag{6}$$

where $E_{ac}$ represents the voltage in $V$ and $R_o$ the internal resistance in $\Omega$.

During discharge $E_{ac}$ is defined as:

$$E_{ac} = E_{\min} + (E_{od} - E_{\min})\frac{q_1}{q_{1\max}} \tag{7}$$

where $E_{\min}$ is the minimum permissible discharge voltage in $V$ and $E_{od}$ the maximum discharge voltage in $V$. The $q_{1\max}$ is defined as:

$$q_{1\max} = cq_{\max} \tag{8}$$

During charge $E_{ac}$ is defined as:

$$E_{ac} = E_{oc} + (E_{\max} - E_{oc})\frac{q_1}{q_{1\max}} \tag{9}$$

where $E_{\max}$ denotes the maximum permitted charge voltage in *V* and $E_{oc}$ the minimum charge voltage in *V*.

The state of charge (*SOC*) of the battery is defined as a linear combination of time and current:

$$SOC_{(t+1)} = SOC_{(t)}(1 - \sigma_{ac}) + I_{ac}ef(\Delta t) \tag{10}$$

where *ef* represents the efficiency factor, $I_{ac}$ the charge/discharge current in A and $\sigma_{ac}$ the discharging rate of the accumulator. $n_{ac}$ and $\sigma_{ac}$ are estimated based on manufacturer data.

### 3.2. KiBaM Experimental Validation

The KiBaM model is experimentally validated for the battery stack described in Section 2. During the experiments, full charge/discharge cycles (depth of discharge = 100%) for various constant currents were implemented. More specifically, the BESS was tested for 15, 20, 25 and 30 A. In Figure 3a the charging voltage curves of the stack for the aforementioned currents are depicted.

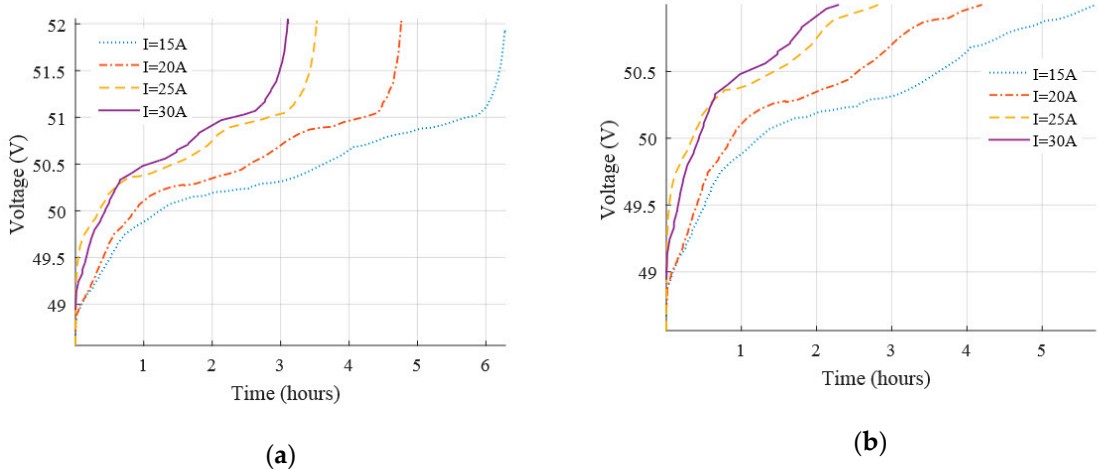

**Figure 3.** Charging voltage curves: (**a**) Whole charging curves; (**b**) Trimmed charging curves.

In the lithium-ion batteries charging curves, an exponential increment of voltage after the voltage reaches the nominal value can be noticed. Similarly, in the lithium-ion batteries discharging curves the voltage decreases exponentially from the maximum voltage to the nominal one. The KiBaM model is developed based on the operation of lead acid batteries [15], in which this phenomenon is not observed. Therefore, the KiBaM model can be applied for the charging/discharging curves of lithium ion batteries only by excluding the exponential area of the curves. Hence, the curves are trimmed in order to apply data fitting to the model (Figure 3b).

As analyzed in Section 2 the residential BESS to be considered consists of 15 lithium iron phosphate battery cells (Model: GBS-LFP100Ah-A) connected in series. The nominal voltage of the GBS-LFP100Ah-A battery is 3.2–3.4 V. This implies that when the cell is fully discharged the nominal voltage is 3.2 V and when is fully charged is 3.4 V. Consequently, the corresponding nominal voltages for the whole stack are 48 V (3.2 V × 15 cells) and 51 V (3.4 V × 15 cells). This is verified by the experimental data (Figure 3a). It is observed at all charging curves, that the initial voltage is 48 V and the initial point of the exponential zone is 51 V.

In order to experimentally validate the KiBaM model, the parameters *c* and *k* which are used in model's equations must be estimated. During the data fitting process, it was found that the optimal values of the aforementioned parameters are $c = 0.045$ and $k = 17$. The data fitting process was applied separately for each cell of the stack and for the whole stack. The performance of the estimated parameters is evaluated by calculating the mean absolute percentage deviation (MAPD) of the four charging curves (Figures 4 and 5). For the charging curve with current I = 15A (Figure 4a) the MAPD

is 0.31%. Additionally, at the charging curve with current I = 20 A (Figure 4b) the MAPD is 0.32%. Moreover, at the charging curves with currents I = 25 A (Figure 5a) and I = 30 A (Figure 5b) the MAPD is 0.29% and 0.34% respectively. Furthermore, by calculating the MAPD separately for the exponential part of the curve and the linear part, is proved that the fitting match is better in the linear part. This is of great significance since the linear part represents the greater part of the curve. More specifically, at the charging curve with current I = 25 A, the MAPD at the exponential part is 0.39% while at the linear part is 0.28%. Moreover, at the charging curve with current I = 30 A, the MAPD at the exponential part is 0.48% while at the linear part is 0.32%.

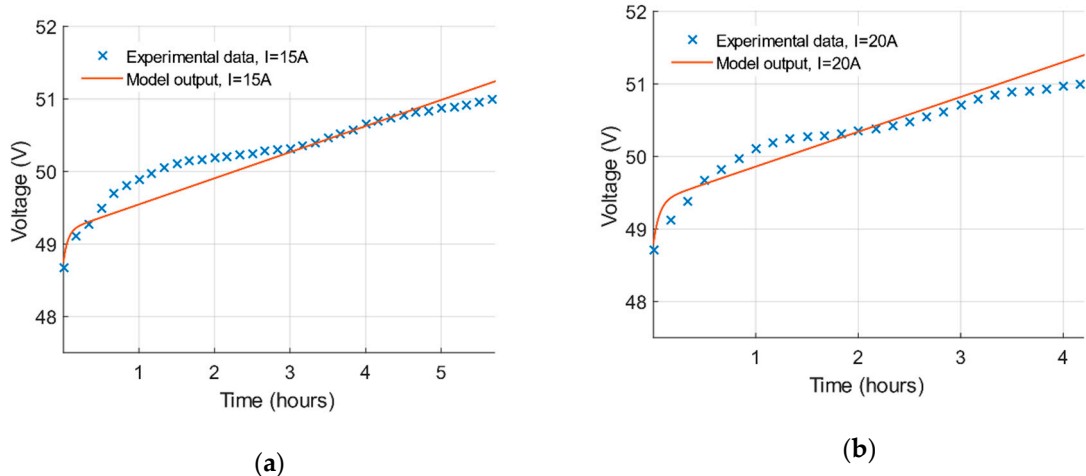

(**a**)

(**b**)

**Figure 4.** Data fitting curves: (**a**) For I = 15 A; (**b**) For I = 20 A.

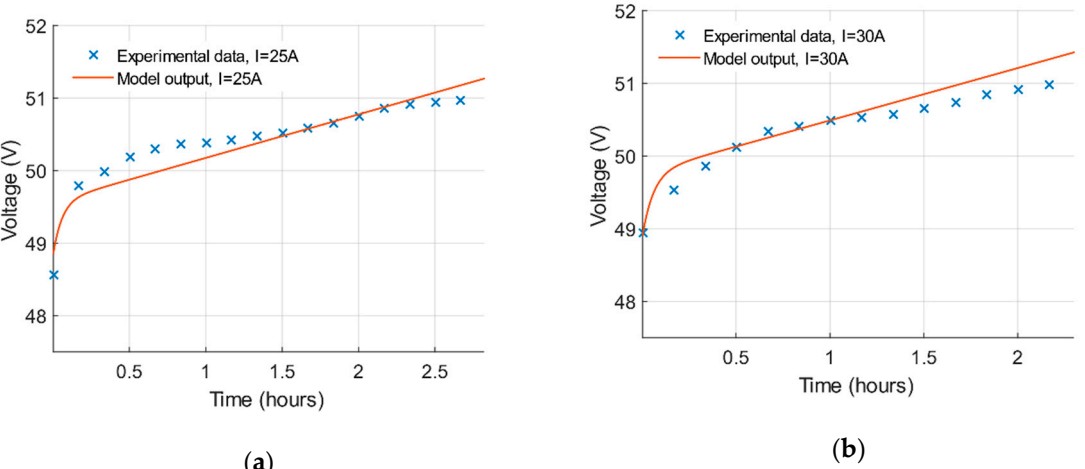

(**a**)

(**b**)

**Figure 5.** Data fitting curves: (**a**) For I = 25 A; (**b**) For I = 30 A.

Over and above, the performance of the estimated parameters is evaluated by calculating the corresponding root mean square error (RMSE) for each charging curve. The RMSE is 0.2 for the 15 A curve, 0.19 for the 20 A curve, 0.18 for the 25 A curve and 0.21 for the 30 A curve, respectively.

### 3.3. Battery Degradation Model

Accurate prediction of lithium-ion batteries' lifetime in any application (residential buildings, factories, stand-alone RES energy stations, cars) is of great importance so as to secure their stability and safety [16,17]. As it is analyzed in [18–20] the lifetime expectancy of a lithium-ion battery can be affected by several factors such as: (a) environmental temperature (*T*), (b) charging/discharging rate (*C_rate*), (c) depth of discharge (*DOD*), (d) number of full charge/discharge cycles and (e) time intervals

between full charge/discharge cycles. In this work, the experimental validated battery ageing model from [21] is utilized. This model comprises the aforementioned ageing factors including the taper voltage of the battery.

As it is thoroughly analyzed in [21,22] the battery ageing model can be stated as an Arrhenius equation. Therefore, the capacity loss is increasing exponentially through time (number of cycles).

$$Q_{loss} = Ae^{\frac{-E_a}{RT}}n^z \tag{11}$$

Equation (11) depends on the temperature and number of cycles. The validation of the model requires the identification of the following battery ageing parameters (Table 1): (a) pre-exponential factor $A$, (b) the activation energy $E_a$ and (c) the cycle's exponent $z$. Those parameters are going to be identified by charging/discharging rate, depth of discharge and taper voltage [21].

**Table 1.** Battery ageing parameters.

| Variable | Description | Unit |
|----------|-------------|------|
| $Q_{loss}$ | Capacity loss | Ah |
| $A$ | Pre-exponential factor | Ah |
| $E_a$ | Activation energy | J*mol$^{-1}$ |
| $R$ | Gas constant | J*mol$^{-1}$K$^{-1}$ |
| $T$ | Temperature | K |
| $n$ | Number of cycles | |
| $z$ | Cycles exponent | |

### 3.3.1. Charging/Discharging Rate

A battery degradation model which comprises the temperature, the number of cycles and the charging/discharging rate is firstly developed in [21]. The capacity loss ($Q_{loss}$) is given by the Arrhenius Equation (12). Cycle's exponent $z$ is set to be constantly 0.74. Furthermore, the Pre-exponential factor $A$ and the activation energy $E_a$ are calculated respectively from Equations (13) and (14) which are obtained as a result of the data fitting process [21]:

$$Q_{loss} = A_{C_{rate}}e^{\frac{-E_{a(C_{rate})}}{RT}}n^{0.74} \tag{12}$$

$$A_{C_{rate}} = e^{\xi_4 C_{rate}^2 - \xi_5 C_{rate} + \xi_6} \tag{13}$$

$$E_{a(C_{rate})} = \xi_1 e^{\xi_2 C_{rate}} + \xi_3 \tag{14}$$

where $\xi_1$, $\xi_2$, $\xi_3$, $\xi_4$, $\xi_5$ and $\xi_6$ symbolize constant parametric coefficients.

### 3.3.2. Depth of Discharge

An ageing model which contains the temperature, the number of cycles and the depth of discharge is developed also in [21]. In this case the charging/discharging rate is considered to be constant. Both activation energy $E_a$ and cycle's exponent $z$ are set to be fixed numbers ($E_a$ = 18,724 and $z$ = 0.74). Pre-exponential factor $A$ is calculated by Equation (16) [21]:

$$Q_{loss} = A_{DOD}e^{\frac{-18724}{RT}}n^{0.74} \tag{15}$$

$$A_{DOD} = \xi_7 + \xi_8 DOD \tag{16}$$

### 3.3.3. Multi Factor Model

The multi-factor model comprises all the degradation factors (environmental temperature, charging/discharging rate, depth of discharge, taper voltage and number of cycles). The cycle's

exponent $z$ is set 0.74. Over and above, activation energy $E_a$ is calculated by Equation (14) and the pre-exponential factor $A$ by Equation (18). Substituting (14) and (18) to (17) yields the final Equation for estimating the capacity loss (19):

$$Q_{loss} = A_{DOD,C_{rate},V_t} e^{\frac{-E_{a(C_{rate})}}{RT}} n^{0.74} \tag{17}$$

$$A_{DOD,C_{rate},V_t} = -\xi_9 + \xi_{10}DOD + \xi_{11}C_{rate} + \xi_{12}e^{\xi_{13}V_t} \tag{18}$$

$$Q_{loss} = \left(-\xi_9 + \xi_{10}DOD + \xi_{11}C_{rate} + \xi_{12}e^{\xi_{13}V_t}\right)e^{\frac{-\xi_1 e^{\xi_2 C_{rate}} - \xi_3}{RT}} n^{0.74} \tag{19}$$

The battery degradation model's parametric coefficients are displayed in Table 2.

**Table 2.** Parametric coefficients.

| $\xi_1$ | $\xi_2$ | $\xi_3$ | $\xi_4$ | $\xi_5$ | $\xi_6$ | $\xi_7$ | $\xi_8$ | $\xi_9$ | $\xi_{10}$ | $\xi_{11}$ | $\xi_{12}$ | $\xi_{13}$ |
|---------|---------|---------|---------|---------|---------|---------|---------|---------|-----------|-----------|-----------|-----------|
| 2330 | 1337 | 13,530 | 433 | 337 | 503 | 2223 | 3138 | 15,767 | 3624 | 1419 | 2721 | 11 |

## 4. Energy Management Strategies in Residential BESS without RES

Since there are two different ToU tariffs to be considered, the EMSs developed are two as well. Both EMSs aim to ensure an economic profit by charging the battery stack from the grid when the energy price is low and discharge the stack to the household's load when the price is high. In order to achieve that the day is segregated into: (a) charging zone(s) and (b) discharging zone(s). During charging zone, the battery stack is charged with constant current. Contrariwise, during the discharging zone the current is determined by an optimizer. The discharging zone is furtherly divided into $N$ time slots. This provides the availability of re-using the optimizer so as to update the optimal discharge schedule of the BESS periodically during the discharging zone.

### 4.1. Energy Management Strategy at Spring/Summer ToU Tariff

During spring and summer, the energy price is 0.078 €/kWh from 11 p.m. to 7 a.m. and 0.11 €/kWh from 7 a.m. to 11 p.m. (Figure 1a). Consequently, at the EMS in spring/summer ToU tariff the day is segregated into one charging zone (11 p.m.–7 a.m.) and one discharging zone (7 a.m.–11 p.m.). This implies that during spring and summer one full charge/discharge cycle per day is permitted. The discharging zone is further divided into $N$ slots. The number of time slots is selected by the end user according to the uncertainty of the forecasted energy consumption profile in the household. For instance, if the BESS is installed in a household where the residents' habits are not changing frequently, the discharging zone could be divided into few timeslots. However, if a statistic analysis proves high alterability of the inhabitant's energy consumption habits the number of timeslots $N$ should be higher. A normal sampling time for updating the optimal discharge schedule at a BESS in a household is between the range of five minutes to one hour. This implies that the discharging zone (16 h) is divided into minimum sixteen timeslots and maximum one hundred ninety-two time slots.

The EMS is displayed in Figure 6 and described as follows. Initially, the end user sets a desirable ageing trajectory for the BESS by utilizing the battery degradation model. Thereupon, the needed data such as the BESS's parameters present values and the day-ahead power demand profile are retrieved. Moreover, the discharging zone is segregated in a desirable number of slots $N$ which is set by the end user. Thereafter, the dynamic optimization algorithm is called in order to define the optimal discharging schedule for the BESS. The optimizer determines the maximum permitted energy supply at the household for each one of the slots. This energy limitation is used also as an instantaneous power limitation at each slot. In that context, the optimizer determines the maximum permitted $DOD$ at each slot and the maximum permissible discharging current as well. At the end of the present time slot, the KiBaM model is called in order to update the status of the BESS based on the actual energy supply and the mean discharge current during the time slot. By taking into account the updated parameters

of the KiBaM model and any possible updates at the forecasted power demand profile the optimizer is recalled so as to define the optimal discharge schedule for the remaining slots of the discharging zone. Once the discharging zone is passed (11 p.m.), the constant charging current is calculated by taking into account the time duration of the charging zone and the SOC of the BESS which is calculated by the KiBaM model. Then, the economic profit for the consumer according to the optimal operation of the BESS through all the day is calculated. The last step of the EMS is to estimate the daily capacity loss of the BESS based on the occurred optimal charging/discharging schedule and the current number of cycle of the BESS.

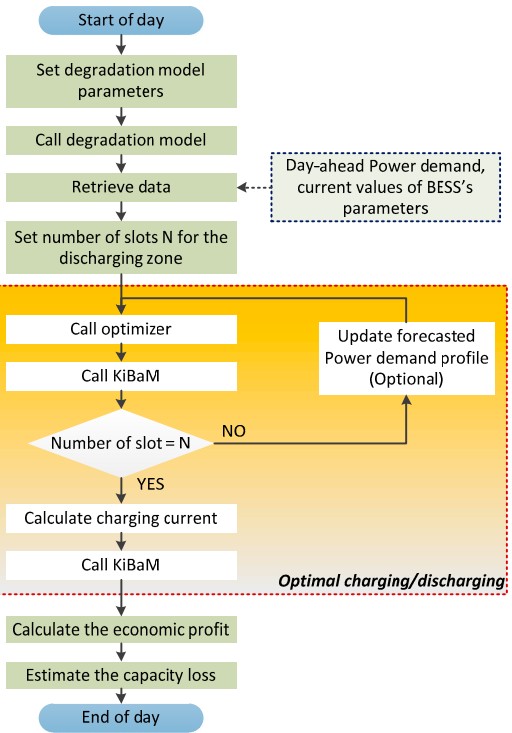

**Figure 6.** EMS at spring/summer ToU tariff.

### 4.2. Energy Management Strategy at Autumn/Winter ToU Tariff

During autumn and winter the energy price is 0.078 €/kWh from 2 a.m. to 8 a.m. and from 3 p.m. to 5 p.m. while the tariff is 0.11 €/kWh from 8 a.m. to 3 p.m. and from 5 p.m. to 2 a.m. (Figure 1b). Therefore, at EMS in autumn/winter ToU tariff the day is segregated into two charging zones (2 a.m.–8 a.m. and 3 p.m.–5 p.m.) and two discharging zones (8 a.m.–3 p.m. and 5 p.m.–2 a.m.). Similarly, with the EMS at spring/summer ToU tariff the discharging zones are further divided into *N* time slots.

The only difference with the previously described EMS is that during autumn and winter the BESS is forced to operate two full charge/discharge cycles through the day in order to make the most of the ToU tariff (Figure 7). The dynamic optimization is firstly called at 8 a.m. and recalled to update the discharge schedule *N* times according to the number of the slots. At 3 p.m. starts the first charging zone which last only two hours. This yields that the BESS might not be fully charged at the beginning of the second discharging zone. Nonetheless, this depends on the desirable ageing trajectory, since the degradation constraint actually regulated how much energy was supplied to the household from the BESS during the first discharging zone. The dynamic optimization algorithm is called again at the beginning of the second discharging zone (5 p.m.) and recalled *N* times so as to update the schedule. Thereafter, the needed charging current is calculated again in order to charge the battery-stack from 2 a.m. to 8 a.m. At the end of the second charging zone the daily economic profit is calculated. Last but not least, the daily capacity loss is estimated by taking into account the operation schedule of the BESS.

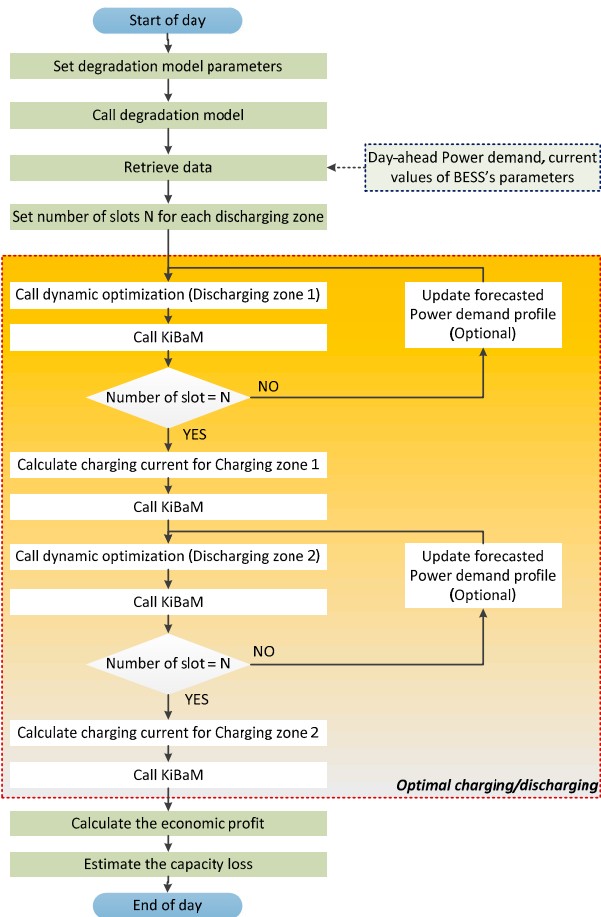

**Figure 7.** EMS at autumn/winter ToU tariff.

### 4.3. Dynamic Optimization

Considering that the actual energy demand might differ from the forecasted demand leads to the notion that applying dynamic optimization at the discharging zones is vital. The optimizer takes under consideration any possible updates on the power demand forecast profile and the present values of the SOC and Voltage of the BESS which are determined by the KiBaM model. The division of the day into charging and discharging zone(s) implies that rolling horizons cannot be used at this dynamic optimization algorithm since this will result to collision between zones of different type (charging and discharging). Thus, the prediction and control horizons are decreased by 1 sample after the end of each slot of the discharging zone. At the beginning of each discharging zone both horizons are reset to the initial value $N$.

The dynamic optimization algorithm defines the optimal discharge profile by utilizing the sequential quadratic programming method. As it is thoroughly analyzed in [23] the non-linear problem is formulated as:

$$\min_{x} F(x) \text{subject to} \begin{cases} G(x) &= 0 \\ H(x) &= \geq 0 \end{cases} \qquad (20)$$

The discretized dynamic model consists of the equality constraints $G(x) = 0$. By setting an initial guess $x_0$ the sequential quadratic programming method iterates are:

$$x_{k+1} = x_k + a_k \Delta x_k, k = 0, 1, \dots \qquad (21)$$

where $\alpha_k$ is the relaxation factor and $\Delta x_k$ is the solution of quadratic programming sub-problem.

$$\min_{\Delta x} \nabla F(x_k)^T \Delta x + \frac{1}{2} \Delta(x)^T A_k \Delta x \tag{22}$$

$$\text{Subject to} \begin{cases} G(x_k) + \nabla G(x_k)^T \Delta x = 0 \\ H(x_k) + \nabla H(x_k)^T \Delta x \geq 0 \end{cases} \tag{23}$$

$A_k$ denotes an approximation of the Hessian of the Lagrangian function:

$$L(x, \lambda, \sigma) = F(x) - \lambda^T G(x) - \sigma^T H(x) \tag{24}$$

The decision variables of the optimization problem are the maximum permitted energy supply at the household from the BESS at the current and upcoming steps of the horizon. The optimizer aims to determine an optimal discharge schedule so as to prevent premature ageing of the BESS. In order to achieve that, a capacity loss constraint is set. The optimizer aims to cover the maximum possible load of the household with respect to the capacity loss constraint. Furthermore, a weight factor $w_i$ is set for each slot of the discharging zone. The weight factor is utilized to prioritize the cover of the load at time slots with low uncertainty regarding the matching of forecasted demand with actual demand. The dynamic optimization problem is stated as:

$$\min_x f(x) = \sum_{i=ts}^{N} \left( E_{d_i} - E_{s_i} \right)^2 - \sum_{i=ts}^{N} \left( w_i E_{s_i} \right)^2 \tag{25}$$

$$\text{s.t.} : E_{s_i} \leq E_{d_i}, i = ts, \dots, N \tag{25a}$$

$$\frac{E_{s_i}}{Q_{nom} \cdot V_i} \leq C_{rate_{max}}, i = ts, \dots, N \tag{25b}$$

$$\sum_{i=ts}^{N} E_{s_i} \leq E_{bat_i}, i = ts, \dots, N \tag{25c}$$

$$SOC_{min} \geq SOC_i - \frac{E_{s_i}}{V_i}, i = ts, \dots, N \tag{25d}$$

$$Q_{loss_i} \leq Q_{loss\_sp}, i = ts, \dots, N \tag{25e}$$

where $E_{d_i}$ symbolizes the energy demand at each step (time slot) of the horizon in kWh, $E_{s_i}$ the maximum permitted energy supply at each step in kWh, $w_i$ the weight factor at each step, $E_{bat_i}$ the available energy at the battery stack at each step in kWh, $V_i$ the battery's voltage at each step in V, $Q_{nom}$ the nominal capacity of the battery in Ah, $C_{rate_{max}}$ the maximum permissible discharge rate, $Q_{loss\_sp}$ the capacity loss set point in Ah, $Q_{loss_i}$ the estimated capacity loss based on the optimal discharge schedule at each step in Ah, $N$ the total number of steps, $ts$ the number of the current step and $i$ the step of the horizon.

Regarding the constraints, first of all the maximum permitted energy supply at each step must be lower or equal to the corresponding energy demand (25a). Additionally, the discharging rate at each step must be lower or equal to the maximum permissible discharging rate ($C_{rate_{max}}$) which is defined by the battery's manufacturer (25b). Furthermore, the overall predicted energy supply for the remaining steps of the horizon must be lower or equal to the present stored energy $E_{bat_i}$ at the BESS (25c). In addition, the SOC must not be reduced lower than the minimum limitation which is set by the end user (25d). Finally, the estimated capacity loss based on the defined discharge schedule must be lower or equal than the capacity loss set point (25e). When the BESS is operating for the very first time ($cycle = 1$), the capacity loss $Q_{loss_i}$ is calculated by substituting (26) and (27) into (19):

$$DOD = \sum_{i=ts}^{N} \frac{100 \cdot E_{s_i}}{Q_{nom} \cdot V_i} \tag{26}$$

$$C_{rate} = \sum_{i\,=\,ts}^{N} \frac{E_{s_i}}{Q_{nom} \cdot V_i} \tag{27}$$

$$Q_{loss_i} = Q_{loss_c} - Q_{loss_{c-1}} \tag{28}$$

where c denotes the total number of cycles including the present cycle.

Afterwards, in order to estimate the capacity loss during the present cycle, Equation (28) is used. This is because the battery degradation model [21] can estimate the capacity loss for a total number of identical cycles. Therefore, in order to estimate the capacity loss only at the present cycle, $c$ identical cycles with the present cycle are considered. Thereupon, the losses for $c$ cycles and $c$-1 cycles are estimated. The subtraction of those two losses (28) denotes the capacity loss for the present cycle. Likewise, the capacity loss set point for each cycle is calculated. Initially, the end user sets *DOD* and $C_{rate}$ at (19) so as to define a desired ageing trajectory. Nonetheless, in order to calculate the capacity loss set point for each cycle at the dynamic optimization problem, (28) is utilized.

### 4.4. End of Day Calculations

At the end of day, the economic profit for the consumer is calculated and the capacity loss of the BESS is estimated. The capacity loss is estimated by taking into account the average current (I) through all the zones of the day (charging and discharging) and the total *DOD*. Over and above, by calculating the cost of charging the BESS with the low tariff (charging zone(s)) and by calculating the cost of reduced load covering from the grid when the tariff is high (discharging zone(s)), the total cost of energy bought from the grid is determined using Equation (29):

$$Cost_{withBESS} = Cost_{chargingBESS} + Cost_{loadcovering} \tag{29}$$

$$Profit = Cost_{withoutBESS} - Cost_{withBESS} \tag{30}$$

Subtracting this cost from the calculated energy cost for the household without a BESS, denotes the economic profit for the consumer (30).

## 5. Analysis of Behavior and Results

The proposed EMSs were implemented in MATLAB for simulation purposes. The simulated scenarios were explored by utilizing data for power consumption in a household in the mainland of Greece (Figure 8) and the aforementioned ToU tariffs (Figure 1a,b).

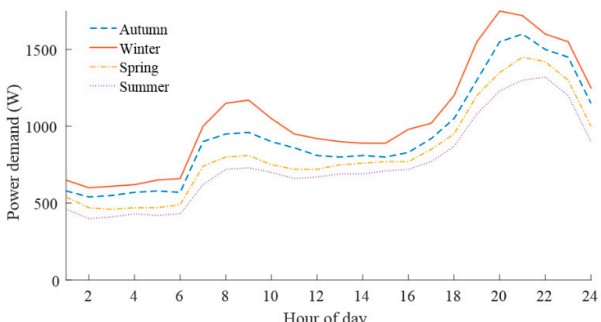

**Figure 8.** Average daily power consumption per season.

### 5.1. Simulation Results at Spring/Summer ToU Tariff

Initially, the EMS at the spring/summer ToU tariff is tested. The degradation model is utilized in order to calculate a desirable ageing trajectory for the battery-stack. By setting to the model the maximum permissible DOD (80%) and Crate (0.33C) which are provided by the battery manufacturer the ageing trajectory is obtained (Figure 9).

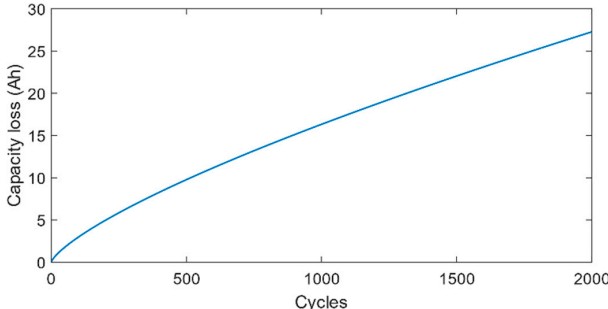

**Figure 9.** Desirable ageing trajectory for the BESS.

The corresponding ageing set point for each step of the horizon is calculated as explained in Section 4.3. Moreover, the sampling time is set to 1 h which implies that the discharging zone is divided into 16 time slots. Furthermore, the maximum permissible $C_{rate}$ is set to 0.33 (25b) and the minimum limitation for the *SOC* (25d) is set to 10%. Last but not least, it is considered that the BESS's state of health is SOH = 100% and thus, the maximum available stored load is 100Ah and the available power is 5.1 kW.

### 5.1.1. Spring Day Scenario

The spring day simulated scenario is implemented by taking under consideration the power consumption profile (Figure 8) for an average day of spring and the ToU tariff of spring/summer (Figure 1a). Additionally, the weights for the 16 time slots of the discharging zone are all set 1. In that context, the optimization algorithm will not prioritize the load cover at any slot of the zone.

Considering that the day-ahead load demand matched the actual load demand Figure 10a is obtained. As it is observed, the EMS distributed the stored energy at the BESS almost equally to the 16 hourly time slots of the discharging zone (around 245 Wh per time slot). This was expected since all 16 weights were set to 1 and there was no mismatch between the forecasted load demand and the actual one. In addition, the overall power supplied to the household was 3979 W and the load at the BESS at the end of the discharging zone was 18 Ah. In order to fully charge the battery stack during the 8 h charging zone the needed current was calculated and found 10.25 A. Figure 10b depicts the discharging (positive currents) and charging currents (negative currents) by starting from the first time slot (7 a.m.) of the discharging zone and ending to the end of the charging zone (7 a.m. of next day).

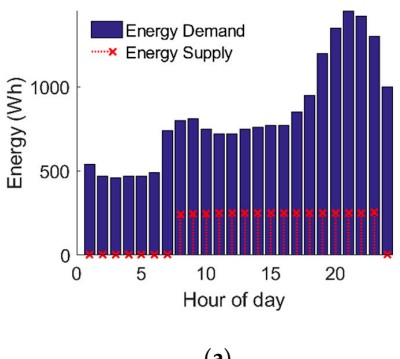

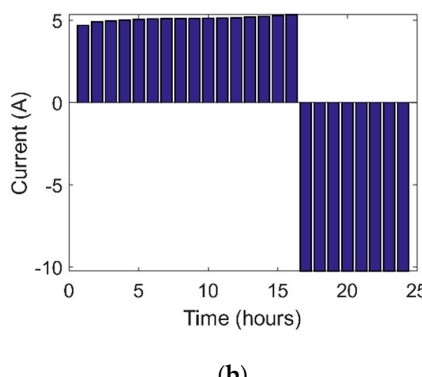

(**a**)                                                (**b**)

**Figure 10.** BESS operation schedule: (**a**) Energy distribution; (**b**) Charging and discharging currents.

Additionally, Figure 11 depicts the BESS parameters which were defined by the KiBaM model by starting from the first time slot of the discharging zone and ending to the end of the charging zone. More specifically is displayed the SOC in %, voltage in V, power in W and load in Ah. Calculating the daily energy cost for the household with and without the BESS it was found that the cost is 1.935 €

with the BESS while it is 2.053 € without the BESS. This implies that the daily profit is 0.118 € and for the whole season would be 10.738 €. The low profit during spring is an outcome of the corresponding ToU tariff which permits only one full charge/discharge cycle per day. Finally, the estimated capacity loss for the whole day is 0.0987 Ah while the capacity loss set point is 0.0984 Ah.

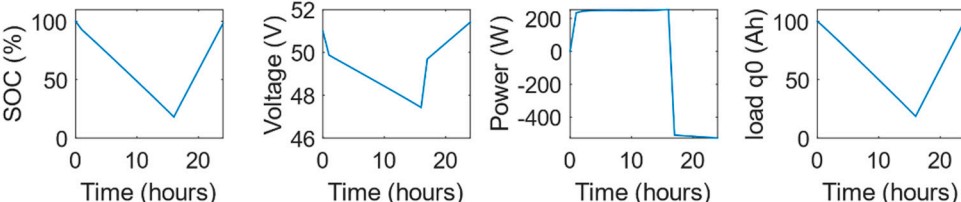

**Figure 11.** BESS parameters.

5.1.2. Summer Day Scenario

Summer day simulated scenario is implemented by taking under consideration the power consumption profile (Figure 8) for an average day of summer and the ToU tariff of spring/summer (Figure 1a). Over and above, the weights for the 16 time slots of the discharging zone are set as: $w_i$ = [1 1 1 1 1 1 1 1 1 1 1 2 2 2 2 2]. In that context, the optimization algorithm will prioritize the load cover at the last five hourly slots of the discharging zone (6 p.m.–11 p.m.).

Considering that the day-ahead load demand matched the actual load demand Figure 12a is obtained. It is observed that the EMS distributed the stored energy just at the slots with low uncertainty for load mismatching (6 p.m.–11 p.m.). More specifically, 807 Wh were supplied from 6 p.m. to 7 p.m., 807.9 Wh from 7 p.m. to 8 p.m., 808.8 Wh from 8 p.m. to 9 p.m., 809.4 Wh from 9 p.m. to 10 p.m. and 809.2 Wh from 10 p.m. to 11 p.m. Additionally, the overall power supplied to the household was 4042 W and the load at the BESS at the end of the discharging zone was 16.08 Ah. In order to fully charge the battery stack during the 8 h charging zone the needed current was calculated and found 10.49 A. Figure 12b depicts the discharging and charging currents by starting from the first time slot (7 a.m.) of the discharging zone and ending to the end of the charging zone (7 a.m. of the next day).

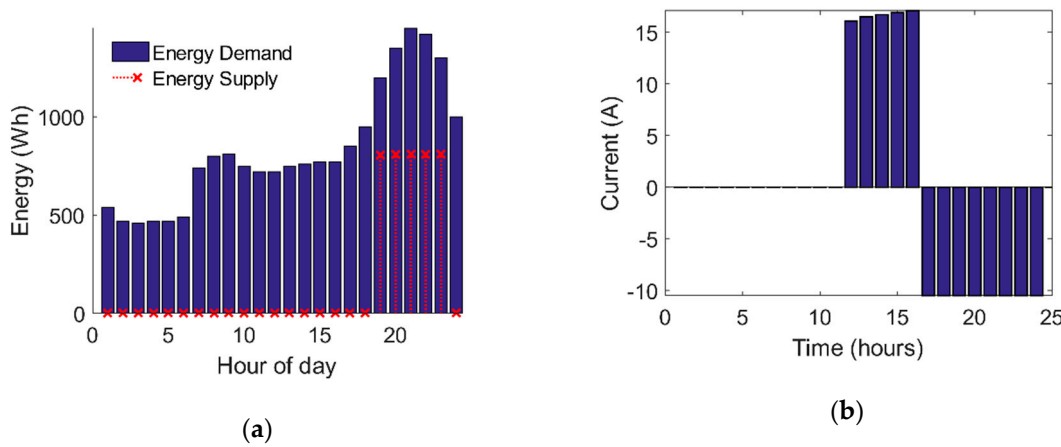

(**a**)

(**b**)

**Figure 12.** BESS operation schedule: (**a**) Energy distribution; (**b**) Charging and discharging currents.

Furthermore, Figure 13 depicts the BESS parameters which were defined by the KiBaM model. Moreover, the daily energy cost for the household with and without the BESS was calculated. With the BESS the cost is 1.746 € while without the BESS is 1.864 €. This implies that the daily profit was 0.118 € and for the whole summer would be 10.738 €. Similarly, with the spring day simulated scenario, the profit for the consumer is not high since the ToU tariff permits only one full charge/discharge cycle per day. Finally, the estimated capacity loss for the whole day is 0.1022 Ah while the capacity loss set point is 0.0984 Ah.

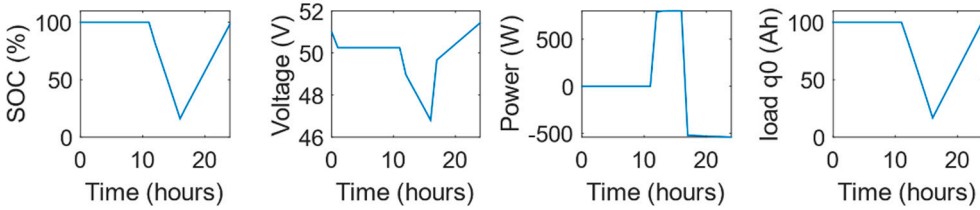

**Figure 13.** BESS parameters.

### 5.2. Simulation Results at Autumn/Winter ToU Tariff

The EMS at the autumn/winter ToU tariff is tested by taking under consideration the same ageing trajectory which was used for testing at spring/summer ToU tariff (Figure 9). In addition, at both discharging zones the sampling time is set to 1 h. Therefore, the first discharging zone is segregated into 5 hourly time slots and the second one into 11 hourly time slots. The maximum permissible $C_{rate}$ is set to 0.33 (25b) and the minimum limitation for the *SOC* (25d) is set to 10%. Last but not least, it is considered that the BESS's state of health is SOH = 100% and thus, the maximum available stored power is 5.1 kW.

#### 5.2.1. Autumn Day Scenario

An autumn day simulated scenario is implemented by taking under consideration the power consumption profile (Figure 8) for an average day of autumn and the ToU tariff of autumn/winter (Figure 1b). This simulated scenario is implemented without prioritizing the load cover at specific slots of the discharging zones. Consequently, all five weights of the first zone and the 11 weights of the second zone are set 1.

The operation of the BESS at this average autumn day is presented at the obtained Figure 14a,b and Figure 15. Due to the equal weights for the all the time slots at the first discharging zone the energy stored is supplied almost equally to the five time slots of the zone.

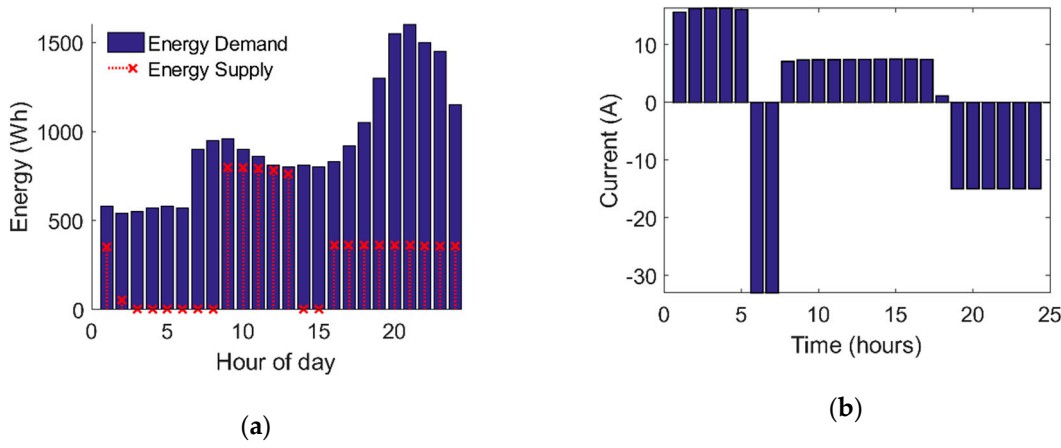

(**a**)

(**b**)

**Figure 14.** BESS operation schedule: (**a**) Energy distribution; (**b**) Charging and discharging currents.

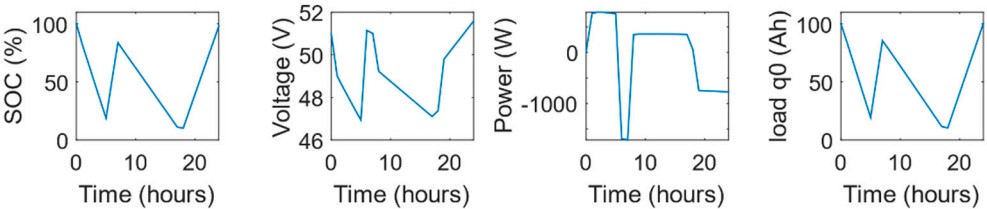

**Figure 15.** BESS parameters.

At the end of first the zone the battery stack's load is 19.45 Ah. Hence, the needed current so as to fully charge the BESS during the short first charging zone (2 h) is 40.28 A. Nonetheless, the BESS $C_{rate}$ limitation (0.33C), forces the implementation of charging with current I = 33 A. Consequently, at the end of the first charging zone the BESS is not fully charged ($q0$ = 85.45 Ah). At the second discharging zone the BESS distributes again the energy stored almost equally to all slots (around 360 Wh) due to the equal weights. However, at the last time slot (1 a.m.–2 a.m.) significantly reduced energy is supplied (52 Wh). This is explained by the fact that the minimum permissible SOC ($SOC_{min}$ = 10%) has been reached. Moreover, the needed current to fully charge the BESS at the second charging zone (6 h) is calculated (I = 15 A) and applied to the KiBaM model.

The daily energy cost for the household with and without the BESS was calculated and found 2.065 € with the BESS and 2.295 € without the BESS. Thus, the daily profit was 0.23 € and for the whole autumn would be 20.93 €. The higher economic profit for the consumer during a typical autumn day compared to a spring/summer day is a result of the corresponding ToU tariff which permits two full charge/discharge cycles per day instead of one. The estimated capacity loss for the first charge/discharge cycle of the day is 0.0864 Ah while the set point is 0.0984 Ah. Additionally, for the second cycle of the day, the capacity loss is 0.0676 Ah and the corresponding set point is 0.0659 Ah. Finally, the estimated capacity loss for the whole day is 0.154 Ah while the capacity loss set point is 0.1643 Ah.

### 5.2.2. Winter Day Scenario

Winter day simulated scenario is implemented by taking under consideration the power consumption profile (Figure 8) for an average day of winter and the ToU tariff of autumn/winter (Figure 1b). At the first discharging zone the weights are set as: $w_i$ = [2 2 1 1 1]. Moreover, at the second discharging zone are as $w_i$ = [1 1 1 2 3 2 2 2 1 1 1]. In that context, during the first discharging zone the EMS will prioritize the load cover at the first two hours (8 a.m.–10 a.m.). In addition, during the second discharging zone, the EMS will prioritize the load cover from 6 p.m. to 11 p.m. with an extra emphasis to the load from 7 p.m. to 8 p.m. (where $w_i$ = 3).

The operation of the BESS at this average winter day is presented at the obtained Figure 16a,b and Figure 17. During the first discharging zone is noticed that from 8 a.m. to 10 a.m. the household's load is fully covered by the BESS while from 10 a.m. to 1 p.m. is covered partially by the BESS and by the grid. At the end of first discharging zone the battery stack's load is 19.23 Ah. In order to fully charge the stack at the first charging zone the needed current is 40.39 A. Nevertheless, due to the BESS $C_{rate}$ limitation (0.33C), the BESS is charged with current I = 33 A. Therefore, at the end of the first charging zone the BESS is not fully charged ($q0$ = 85.23 Ah). During the second discharging zone, the BESS targets to fully cover the load from 7 p.m. to 8 p.m. (where $w_i$ = 3) by discharging the stack to the house's load with I = 33 A. However, due to the $C_{rate}$ limitation is achieved to cover 1612 Wh of the overall 1750 Wh. The rest 138 Wh are covered from the grid. Additionally, the BESS partially covers the loads from 6 p.m. to 7 p.m. (supplied energy is 581.8 Wh), from 8 p.m. to 9 p.m. (supplied energy is 591.6 Wh), from 9 p.m. to 10 p.m. (supplied energy is 591 Wh), from 10 p.m. to 11 p.m. (supplied energy is 227.6.8 Wh) and from 11 p.m. to midnight (supplied energy is 11.1 Wh). From 1 p.m. to 6 p.m. and from midnight to 2 a.m. the household's load if fully covered by the grid.

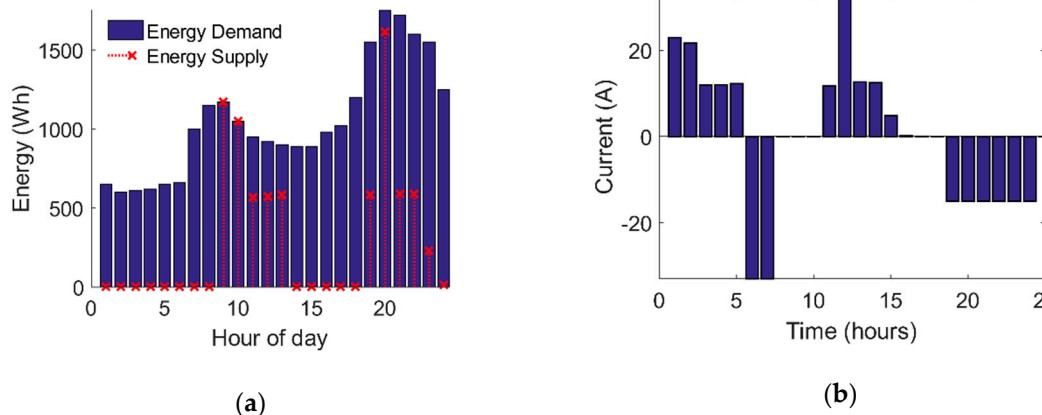

**Figure 16.** BESS operation schedule: (**a**) Energy distribution; (**b**) Charging and discharging currents.

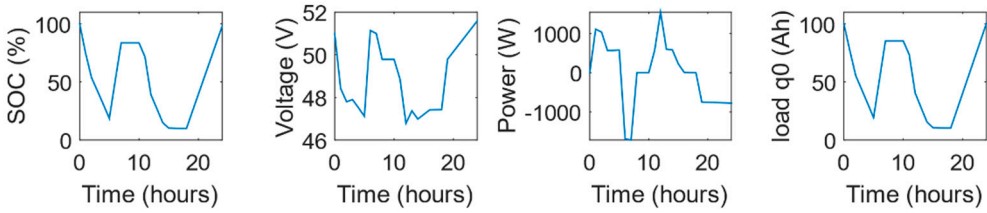

**Figure 17.** BESS parameters.

At the end of the second discharging zone it is noted that the BESS SOC is 10%, which implies that the minimum limit ($SOC_{min}$) was reached. BESS's load ($q0$) was reduced by 75.23 Ah and the average discharging current was I = 6.82 A ($C_{rate}$ = 0.0682). This yields that the estimated capacity loss at the optimal discharge schedule is probably significantly lower than the corresponding set point (25e) which was calculated for DOD = 80% and $C_{rate}$ = 0.33. Hence, the optimizer could possibly permit further discharging of the BESS. Nonetheless, for safety reasons, further discharging was prevented from constraint (25d). At the second charging zone the BESS is charged with current I = 15 A in order to fully charge the battery stack (Figure 10b).

Calculating the daily energy cost for the household with and without the BESS, it was found that the cost is 2.351 € with the BESS and 2.579 € without the BESS. This implies that the daily profit is 0.228 € and for the whole winter would be 20.75 €. The increased profit during a typical winter day compared to a spring/summer day is an outcome of the corresponding ToU tariff which permits two full charge/discharge cycles per day instead of one. The estimated capacity loss for the first charge/discharge cycle of the day is 0.0859 Ah while the set point is 0.0984 Ah. In addition, for the second cycle of the day, the capacity loss is 0.0681 Ah and the corresponding set point is 0.0659 Ah. Finally, the estimated capacity loss for the whole day is 0.154 Ah while the capacity loss set point is 0.1643 Ah.

### 5.3. Multi-Season Simulations

The above simulated scenarios demonstrated the average daily operation of the BESS at the four seasons of the year. In all four simulated scenarios the capacity loss set points were calculated by utilizing the battery manufacturer's hard limits (DOD = 80% and Crate = 0.33). By running multi-season simulations, the BESS's capacity fade is estimated. In addition, the total economic profit for the consumer and the economic profit per season is calculated. Over and above, further multi-season simulations were implemented by using different ageing trajectories for the BESS. The main purpose of this task, was to explore the BESS's behavior when stricter restrictions are applied regarding the ageing prevention and how the economic profit is affected.

The simulated scenarios were implemented for five different capacity loss trajectories. The trajectories were defined for the following DODs: (a) DOD = 80%, (b) DOD = 70%, (c) DOD = 60%, (d) DOD = 50% and (e) DOD = 40%. At all five trajectories the Crate was set at its maximum permitted value (0.33). The duration of all simulated scenarios is 1 year. Considering that during autumn/winter the BESS is forced to operate two full charge/discharge cycles per day and during spring/summer one cycle per day the overall number of cycles through one year is 547. The estimated capacity loss for each scenario and the corresponding capacity loss set points are depicted in Figure 18.

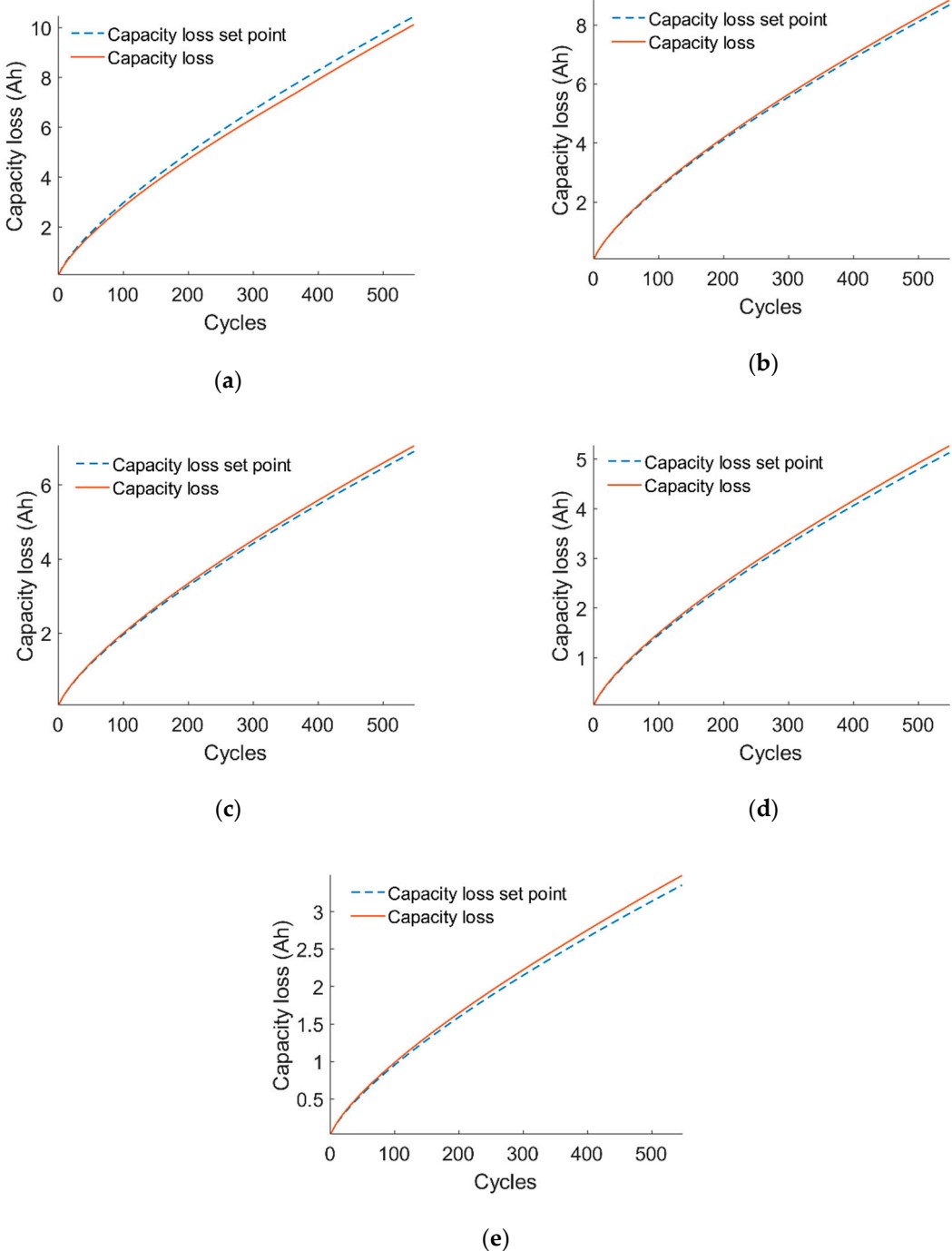

**Figure 18.** BESS estimated capacity loss and capacity loss trajectory: (**a**) for DOD = 80%; (**b**) for DOD = 70%; (**c**) for DOD = 60%; (**d**) for DOD = 50%; (**e**) for DOD = 40%.

Excluding the first scenario where the ageing trajectory is determined for DOD = 80%, at all the other scenarios the estimated capacity loss at the end of the year is slightly higher than the corresponding set point. This is an outcome of applying dynamic optimization only at the discharging zones. The non-constrained operation during charging zones, increases the average operating current of the BESS and hence the estimated capacity loss. Nevertheless, applying charging without considering ageing constraints is inevitable in order to store the maximum possible energy at the BESS and hence achieve higher economic profit.

During the one-year simulation for the 80% DOD ageing trajectory the economic profit was found to be 63.42 € and the estimated capacity loss 10.12 Ah. Considering that the end of lifetime of a battery is when the capacity has been reduced to 80% of the nominal capacity, the BESS has surpassed half of the expected lifetime after just one year of operation. By replacing the ageing trajectory with the trajectory for DOD = 70%, the profit is reduced to 57.41 € and the capacity loss to 8.84 Ah. This implies that the profit has been reduced by 9.5% while the capacity loss has been decreased by 12.6%. Therefore, by applying a slightly stricter ageing constraint to the dynamic optimization, it is observed that the premature ageing prevention rate is greater than the rate of the decreased economic profit. By applying a stricter capacity loss constraint (ageing trajectory DOD = 60%), it is noted that the economic profit after 1-year operation is reduced 48.57 € and the battery stack's capacity fade to 7.05 Ah. Compared to the corresponding values of the first simulated scenario (trajectory for DOD = 80%) the economic profit is decreased by 23.4% while the BESS's capacity loss is reduced by 30.3%. Over and above, by applying the ageing trajectory for DOD = 50% the profit is furtherly decreased to 40.79 € (35.7% decrement compared to scenario DOD = 80%) and the capacity loss to 5.26 Ah (48% decrement compared to scenario DOD = 80%). Finally, at the last simulated scenario is applied the relative ageing trajectory for DOD = 40%, and is observed that the profit is reduced to 32.34 € (49% decrement compared to scenario DOD = 80%) while the capacity loss is decreased to 3.48 Ah (65.6% decrement compared to scenario DOD = 80%).

The obtained results (Table 3) of the above multi-season simulated scenarios prove two significant conclusions. First of all, the DOD is a massive impact factor in the battery's capacity fade. Reducing the DOD to half (from 80% to 40%) results in a 65.6% reduction of the capacity loss of the BESS. Secondly, it is possible to significantly increase the economic profit in the long-term by preventing premature ageing of the BESS. Especially, when a BESS operates on a ToU pricing environment where the opportunities for economic profit are few (compared to RTP tariffs) and prefixed, the only way to increase profit is by extending the lifetime expectancy of the BESS.

**Table 3.** Multi-season simulated scenario's results.

| DOD | Profit during Autumn | Profit during Winter | Profit during Spring | Profit during Summer | Overall Profit (1 Year) | Capacity Loss (1 Year) |
|-----|-----|-----|-----|-----|-----|-----|
| 80% | 20.9 € | 20.9 € | 10.75 € | 10.87 € | 63.42 € | 10.12 Ah |
| 70% | 19.25 € | 19.25 € | 9.38 € | 9.53 € | 57.41 € | 8.84 Ah |
| 60% | 16.25 € | 16.25 € | 7.99 € | 8.08 € | 48.57 € | 7.05 Ah |
| 50% | 13.78 € | 13.78 € | 6.58 € | 6.65 € | 40.79 € | 5.26 Ah |
| 40% | 10.98 € | 10.98 € | 5.16 € | 5.22 € | 32.34 € | 3.48 Ah |

## 6. Conclusions

This work demonstrated the development of two EMSs for a residential BESS. Both strategies are designed for application on a BESS which consists of 15 lithium iron phosphate battery cells and operates without RES at the Greek ToU pricing environment. In addition, both of the strategies utilize the KiBaM model which is validated for the aforementioned BESS and a validated battery ageing model for lithium iron phosphate batteries.

The EMSs are designed to take advantage of the seasonal Greek ToU tariff so as to reduce the electricity costs at the household. This is achieved by segregating the day into charging and discharging

zone(s). During the discharging zones the BESS's operation is determined by a dynamic optimization algorithm in order to efficiently discharge the battery stack at the household. The optimization algorithm has the capability of prioritizing the load coverage at specific times of the discharging zone when there is low uncertainty for load mismatching with the forecasted load demand. Over and above, the optimizer takes into account various physical constraints and a capacity loss trajectory which can be adjusted by the BESS's end user. On the contrary, the charging is applied without considering any constraints in order to achieve to fully charge the BESS if possible, with energy when the tariff is low. This will result in a higher load coverage from the BESS when the tariff is high.

The simulation results illustrate the ability of the EMSs to achieve high economic profits with respect to the constraints set. The daily economic profit for the consumer is not notably high during spring and summer days. This is due to the corresponding ToU tariff which permits only one full charge/discharge cycle per day. Contrariwise, during autumn and winter days a significantly increased economic profit is observed because of the tariff which permits the operation of two full charge/discharge cycles per day. Nonetheless, a problem to be considered during autumn/winter is that the first charging zone is remarkably short (2 h) and therefore in many cases the BESS is not fully charged at the beginning of the day's second discharging zone. This evinces the significance of distributing the energy stored efficiently by supplying energy at times with low uncertainty for load mismatching. Consequently, the EMSs performance can significantly improve when the ToU tariff comprises many fare alterations and when it takes under consideration forecast uncertainty.

Furthermore, the multi-season simulated scenarios illustrate that EMSs manage to operate the BESS by following a specific ageing trajectory. Above all, is proved that it is possible to increase the economic profit by extending the lifetime expectancy of the BESS. This is of great importance, considering that nowadays BESSs are commonly used with RES or without RES in RTP tariffs. Nevertheless, by applying strict restrictions regarding the BESS's capacity loss, it is observed that the economic profit on the long term can be increased notably. The simulation results proved that by reducing the DOD for calculating the ageing trajectory from 80% to 40% the capacity loss of the battery was decreased 65.6% while the economic profit was reduced by 49%. Consequently, when there is not a RES or RTP tariff, preventing premature ageing at a BESS is crucial in order to maximize the economic profit.

**Author Contributions:** Conceptualization, S.P.; methodology, C.G.; software, C.G.; validation, C.Z.; formal analysis, C.G. and C.Z.; investigation, C.G.; data curation, D.T.; writing—original draft preparation, C.G.; writing—review and editing, C.Z.; supervision, S.P.; project administration, S.V.; funding acquisition, C.Z., S.P. and S.V. All authors have read and agreed to the published version of the manuscript.

**Funding:** This research received no external funding.

**Acknowledgments:** Research supported by EU funded HORIZON2020 project inteGRIDy-integrated Smart GRID Cross-Functional Solutions for Optimized Synergetic Energy Distribution, Utilization & Storage Technologies, H2020 Grant Agreement Number: 731268.

**Conflicts of Interest:** The authors declare no conflict of interest.

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
