# Peer review of "Optimal Operation of a Residential Battery Energy Storage System in a Time-of-Use Pricing Environment"

_applsci, doi:10.3390/app10175997_

Round 1

Reviewer 1 Report

The battery aging investigation is as well as battery modification is a hot topic currently. Using Energy Management Strategies is necessary for prolongation of battery durability. Authors have done extensive optimisation considering price of electricity based on modelling.
I recommend to publish this article after minor corrections:
1) Line 102 Owing to the absence of RES, the BESS is charged from the grid when the tariff is 0.78 €/kWh and discharges to the house when the tariff is 0.11 €/kWh. May be 0.078 €/kWh
2) Line 186 It is not understandable which kind of fitting was done. The number of 99.69% based on observation of Figure 4 (experimental data and calculated) looks doubtful. I would kindly ask more detailed information. Also based on other figures 4 a and b, 5 a and b
3) Figure 8 Y-axis - power demand

Reviewer 2 Report

The optimization model does not consider the cost of the BESS. If BESS needs to be replaced, there should be a cost.

In the dynamic optimization model in (25), does it assume the energy demand stay as a constant during each time slot?

How many time slots are considered in the optimization model? For 10 years?

In the abstract, it is said that “the objective of the developed EMSs is to reduce capacity loss without over-affecting the BESS’s performance regarding load covering”. But the optimization model does contain any BESS protection to prevent BESS.

n represents several different symbols in this paper.

Reviewer 3 Report

The article is interesting, but the model presented in Figure 4 is not well fitted.

Author Response

Reviewer 3

  • The article is interesting, but the model presented in figure 4 is not well fitted.

Response to reviewer 3

Dear reviewer, thank you very much for your comment. We would like to kindly inform you that:

  • In order to define the Kinetic Battery Model’s parameters k and c, non-linear regression was applied in Matlab by utilizing the function nlinfit. In the revised version of the article, the mean absolute percentage deviation (MAPD) was applied separately for the exponential and the linear part of the curve. It is observed that the MAPD is lower in the linear part. Therefore, the fitting was considered acceptable since the linear part represents the greater part of the curve that we are interested in working. Additionally, even in the worst fitted parts of the curve the deviation is around 0.4-0.5 V which implies that the fitting mismatch is ~1% considering that the BESS’s voltage range is up to 48-51V. Therefore, we considered that this deviation was not that impactful and we decided to use the presented model with the defined parameters.

As a future work we will consider an enhanced version of the model that will be able to better capture the exponential part as well by considering a more detailed approach. Thank you for providing this comment.

Round 2

Reviewer 2 Report

The revised paper is OK for the reviewer.